# Synergistic Effects of Earthworms and Plants on Chromium Removal from Acidic and Alkaline Soils: Biological Responses and Implications

**DOI:** 10.3390/biology12060831

**Published:** 2023-06-08

**Authors:** Ping Liu, Yan Song, Jie Wei, Wei Mao, Jing Ju, Shengyang Zheng, Haitao Zhao

**Affiliations:** 1College of Environmental Science and Engineering, Yangzhou University, Yangzhou 225127, China; liuping329098@163.com (P.L.); 007816@yzu.edu.cn (Y.S.); xsqwinds@163.com (J.W.); maowei918@163.com (W.M.); jujing@yzu.edu.cn (J.J.); syzheng@yzu.edu.cn (S.Z.); 2Key Laboratory of Cultivated Land Quality Monitoring and Evaluation, Yangzhou University, Ministry of Agriculture and Rural Affairs, Yangzhou 225127, China

**Keywords:** synergistic effects, earthworms, plant, chromium removal, biological responses

## Abstract

**Simple Summary:**

This paper focuses on the role of synergistic removal effects of earthworms and plants on two types of chromium-contaminated soils and their biological responses. We firstly examined the synergistic removal rate of chromium from contaminated acidic and alkaline soils via earthworms and plants, and then analyzed the speciation characteristics of chromium in soil after the experiment. Additionally, we also explored the accumulation of chromium in organisms and the earthworm intestinal bacterial community to reveal the biological responses. The main contribution of this study was to successfully prove that the combined use of *Eisenia fetida* and ryegrass had the best effect on removing chromium pollution from soil, and acidic soil exerted stronger pressure on earthworms which finally unbalanced the intestinal bacterial phyla. In future studies, we should consider not only the removal effect of heavy metals on soil via organisms, but also the stress effect on organisms themselves, and weigh up the advantages and disadvantages, which would be more conducive to the sustainable development of the ecological environment.

**Abstract:**

Soil heavy metal pollution has become one of the major environmental issues of global concern and solving this problem is a major scientific and technological need for today’s socio-economic development. Environmentally friendly bioremediation methods are currently the most commonly used for soil heavy metal pollution remediation. Via controlled experiments, the removal characteristics of chromium from contaminated soil were studied using earthworms (*Eisenia fetida* and *Pheretima guillelmi*) and plants (ryegrass and maize) at different chromium concentrations (15 mg/kg and 50 mg/kg) in acidic and alkaline soils. The effects of chromium contamination on biomass, chromium bioaccumulation, and earthworm gut microbial communities were also analyzed. The results showed that *E. fetida* had a relatively stronger ability to remove chromium from acidic and alkaline soil than *P. guillelmi*, and ryegrass had a significantly better ability to remove chromium from acidic and alkaline soil than maize. The combined use of *E. fetida* and ryegrass showed the best effect of removing chromium from contaminated soils, wih the highest removal rate (63.23%) in acidic soil at low Cr concentrations. After soil ingestion by earthworms, the content of stable chromium (residual and oxidizable forms) in the soil decreased significantly, while the content of active chromium (acid-extractable and reducible forms) increased significantly, thus promoting the enrichment of chromium in plants. The diversity in gut bacterial communities in earthworms decreased significantly following the ingestion of chromium-polluted soil, and their composition differences were significantly correlated with soil acidity and alkalinity. *Bacillales*, *Chryseobacterium*, and *Citrobacter* may have strong abilities to resist chromium and enhance chromium activity in acidic and alkaline soils. There was also a significant correlation between changes in enzyme activity in earthworms and their gut bacterial communities. The bacterial communities, including *Pseudomonas* and *Verminephrobacter*, were closely related to the bioavailability of chromium in soil and the degree of chromium stress in earthworms. This study provides insights into the differences in bioremediation for chromium-contaminated soils with different properties and its biological responses.

## 1. Introduction

In modern industrialized societies, chromium is widely used in various fields, including the leather industry, electroplating, and steel production. The indiscriminate discharge of chromium-containing pollutants into the environment has a serious impact on the atmosphere, water bodies, and soil [1,2]. Additionally, chromium is also considered to be a carcinogenic substance [3], and long-term exposure to chromium-polluted environments may lead to health problems such as asthma, dermatitis, and organ dysfunction [4,5]. Therefore, the remediation and management of chromium-polluted soil are necessary.

There are many studies on the remediation of heavy-metal-contaminated soils using different methods, such as chemical, physical, and biological remediation. Although these methods have been quite effective, each method has its own advantages and disadvantages [6]. For example, chemical and physical approaches reduce the activity and hazard risk of heavy metals via passivation, but the cost is high, and these methods may cause secondary pollution [7]. Compared with methods such as chemical and physical remediation, biological remediation has advantages such as being economical, environmentally friendly, and sustainable, and can effectively protect the soil. Therefore, biological remediation has received increasing attention and application globally [8,9].

Plant resources for biological remediation are scarce, grow slowly, and are limited to certain hyperaccumulator plants, such as ryegrass, maize, and *Solanum nigrum* L. [10,11,12]. Animal remediation involves using soil animals (such as earthworms and nematodes) to absorb, degrade, or transfer heavy metals, thereby reducing the concentration of heavy metals in soil [13,14]. As “engineers” of terrestrial ecosystems, earthworms play a crucial role in maintaining soil structure, air permeability, and stability [15]. Earthworms can change soil properties through their life activities, including feeding, burrowing, and excreting, thereby increasing the bioavailability of soil heavy metals and accelerating the plant remediation of heavy metals [16,17,18]. For example, changes in soil pH and organic matter content can influence the occurrence and final bioavailability of heavy metals [19,20]. In addition, earthworms can significantly enrich heavy metals in their bodies. Van Hook et al. found that earthworms fed chromium-polluted soil had an enrichment effect, and the chromium content in their bodies was 17 times that of the soil [21]. However, studies have shown that different ecological types of earthworms have different abilities to accumulate heavy metals in their bodies [22].

Each animal species has its own tolerance range for heavy metals in polluted soil, and once this tolerance range is exceeded, the animals will attempt to leave the polluted area or may die [23]. When animals accumulate heavy metals, there is a trade-off between remediation effectiveness and organism health. Recent reports have shown that the ingestion of heavy metals, silver nanoparticles, microplastics, and antibiotics affects the composition of the host gut microbial communities, such as those in mice, collembolans, and honeybees [24,25,26,27]. Therefore, abnormal reactions of the gut microbial communities can provide a new way for characterizing the toxicity of pollutants. However, the potential impact of chromium-polluted soil on the gut microbial communities of earthworms is not clear.

To date, most studies have focused on the individual effects of earthworms or plants on the removal of chromium from soil. Since a single remediation technique has significant limitations, the combination of two or more different biological remediation technologies and methods has become an important research direction for soil pollution remediation [28,29]. This study used combinational treatment approaches with different ecological types of earthworms and different types of plants for the removal of chromium and its retention form in acidic and alkaline soil. The biological response characteristics of earthworms and plants to chromium were also analyzed, providing a theoretical basis and practical reference for the synergistic biological removal of heavy metals from soil [30,31].

## 2. Materials and Methods

### 2.1. Soil, Earthworms, and Other Materials

To investigate the effectiveness of earthworm–plant synergy in removing chromium from soil, we collected acidic and alkaline soils from Sanming City, Fujian Province (South China) and Yancheng City, Jiangsu Province (mid-eastern coast of China), respectively, for this experiment. The physicochemical properties of the soils are shown in Table 1, and the sampling areas were free from ground cover or known contamination. After removing all of the plant roots and impurities, the soil samples were air-dried at room temperature (22–26 °C) and sieved through a 2 mm screen. For the experiment, 1-month-old *Eisenia fetida* (200–300 mg) and *Pheretima guillelmi* (0.8–1.2 g) were purchased from a Wang Jun earthworm farm in Jurong, Jiangsu Province, China. These earthworms were characterized by high vigor and sensitivity to external stimuli. Prior to the experiment, all of the earthworms were acclimated to the test soil for 2 weeks under laboratory conditions and were well-developed and active. The experiment also used ryegrass (Dongmu 70) and maize (Zhongnong Sweet 488) purchased from Yangzhou Seed Company (Yangzhou, China). Orange-red crystalline pellets of potassium dichromate (K_2_Cr_2_O_7_) were purchased from Shanghai Sinopharm Chemical Reagent Co. (Shanghai, China).

### 2.2. Experimental Design

Based on preparatory experiments, earthworms began to escape when the chromium concentration in the soil was greater than 60 mg/kg. To investigate the effect of earthworms in synergy with plants on the removal of chromium from soil, acidic and alkaline soils with different concentrations of chromium (low, L: 15 mg/kg; high, H: 50 mg/kg) were used and we introduced three variables each for earthworm and plant inoculation, earthworm inoculation (e0: no earthworm inoculation, E: *E. fetida* inoculation, and P: *P. guillelmi* inoculation), and plant cultivation (p0: no plant cultivation, R: ryegrass cultivation, and C: maize cultivation). A total of 36 treatments were set up with three replicates per treatment, as shown in Table 2.

A 30 g/L solution of potassium dichromate was added to both soil types and stirred evenly to prepare the contaminated soil with high concentrations of chromium (50 mg/kg), while the original chromium concentration in the soils was considered to be the low chromium concentration (approximately 15 mg/kg). The soil was then left to stand for 2 weeks to allow the chromium concentration to stabilize at either low or high levels. During the experiment, 2 kg of contaminated soil was placed in plastic pots (diameter: 16 cm, bottom diameter: 13 cm, height: 17.5 cm) and planted with either 100 ryegrass seeds or six maize seeds. In addition, 40 *E. fetida* or 20 *P. guillelmi* were added to the pots that needed to be inoculated with earthworms. The pots were then placed under natural indoor conditions with a temperature range of 20–26 °C during the day and 16–20 °C at night. Pure water was added periodically using a weighing method to maintain consistent soil moisture. After 30 days of growth, the soil, plant, and earthworm samples were collected, and various parameters were measured.

### 2.3. Cr in Soil

Soil pH was measured using a pH meter and a 5:1 water-to-soil ratio extraction method [32]. Soil organic matter was determined using the loss-on-ignition method and a muffle furnace [33]. Total chromium and various forms of chromium in the soil were measured. Total chromium in the soil was determined using an acid digestion method [34]. Dried and ground soil samples (1 g) were weighed using a precision balance (0.0001 g) and digested and extracted with a mixture of concentrated nitric acid and hydrochloric acid (50%) at 95 °C. The samples were filtered through Grade 1 filter paper and stored in containers for analysis. To assess possible errors, control samples were prepared for each series of samples [35]. The determination of acid-extractable Cr, reducible Cr, oxidizable Cr, and residual Cr in the soil was performed using the Tessier sequential extraction method [36]. The percentage of each form of Cr content was calculated as the proportion of the total Cr content (%): (content of each form of Cr/total Cr content) × 100. The removal rate of Cr from the soil was calculated as follows (%): (the content of total Cr in the soil before the experiment—the content of total Cr in the soil after the experiment)/the content of total Cr in the soil before the experiment × 100.

### 2.4. Biological Responses of Earthworms to Cr Exposure

At the end of the experimental period, earthworm samples were manually separated from the soil using forceps. The mechanical manipulation was kept to a minimum to avoid damage to the earthworms, and the earthworms were then placed on qualitative filter paper soaked in pure water for 12 h to remove intestinal contents. To measure the concentration of Cr in the earthworms, the samples were frozen at −30 °C and then dried in an oven at 65 °C for 48 h [37]. The dried samples were ground and placed in covered vials. The Cr was measured using an acid digestion method [38]. In this method, 0.5 g of earthworm tissue was weighed, added to a test tube with 5 mL of concentrated nitric acid and 1 mL of hydrogen peroxide, and heated at 180–220 °C until a clear solution was obtained. The samples were filtered after cooling.

The concentration of Cr accumulated in the earthworms (mg/kg) was calculated as follows: (dry weight of earthworm samples after the experiment × concentration of Cr in the earthworms). To determine the toxic effects of Cr contamination on the earthworms, five healthy earthworms were collected from every treatment, and their mitochondrial reactive oxygen species (ROS), superoxide dismutase (SOD), catalase (CAT), glutathione S-transferase (GST), and malondialdehyde (MDA) activities were detected using a kit from Suzhou Keming Biotechnology Co., Ltd. (Suzhou, China).

Additionally, earthworms of similar sizes were selected from every treatment, washed with distilled water, and rapidly frozen using liquid nitrogen for microbial community analysis. The contents were dissected under sterile conditions, collected in sterile tubes, and then stored at −80 °C for sequencing analysis. DNA was extracted from the earthworm gut (0.5 g) using the FastDNA SPIN Kit (MP Biomedicals, Santa Ana, CA, USA), fragmented via mechanical interruption (ultrasound), and then the fragmented DNA was purified, end-repaired, 3′-end added, and had a sequencing junction attached. The fragment size was selected via agarose gel electrophoresis, and PCR amplification was performed to form sequencing libraries. The constructed libraries were first subjected to library quality control, and the libraries that passed the quality control were then sequenced using the Illumina sequencing platform according to the standard protocol provided by Shanghai Majorbio Biopharm Technology Co., Ltd. (Shanghai, China).

### 2.5. Biological Responses of Plants to Cr Exposure

When the plants were mature, the above-ground parts of the plants were harvested, washed with deionized water, blanched at 105 °C for 0.5 h, and dried at 70 °C until a constant weight was reached. Following this, the dried plant tissue samples were digested using the HNO_3_-H_2_O_2_ digestion method, and the Cr concentration in the test solution was measured using flame atomic absorption spectrometry [39]. The Cr accumulation in the above-ground parts of the plants (mg/kg) was calculated as the product of the dry weight of the above-ground parts of the plant after the experiment and the concentration of Cr in them.

### 2.6. Statistical Analyses

The data were analyzed using Excel software and SPSS Statistics 23. The normal distribution of the data was assessed using the Shapiro–Wilk test and transformed accordingly. One-way analysis of variance (ANOVA) was used to analyze the data, and Duncan analysis was performed to determine the significance of differences between treatments. General plotting was completed using Origin 23, and the Majorbio platform was used to generate Venn and Circos plots of the microbial communities. A *p*-value less than 0.05 was considered to be statistically significant.

## 3. Results

### 3.1. Removal and Speciation Characteristics of Chromium in Soil

In acidic and alkaline soils, planting ryegrass alone was more effective in removing chromium from the soil compared to planting maize alone (Figure 1). *E. fetida* was more effective in removing chromium than *P. guillelmi*, and the combination of *E. fetida* and ryegrass showed the best performance in acidic soil with a low chromium concentration (removal rate of 62.64%). However, in alkaline soil, the combination of earthworms and maize did not significantly improve the removal rate of chromium compared to earthworms alone (Figure 1B).

The addition of earthworms and plants significantly affected the soil pH (Figure 2). In acidic soil, *E. fetida* contributed more to the removal of chromium from the soil than *P. guillelmi*, and the soil pH was the highest in the combination treatment of earthworms and ryegrass (pH 6.16 ± 0.05) (Figure 2A). In alkaline soil, *E. fetida* caused a greater decrease in soil pH than *P. guillelmi*, and ryegrass caused a greater decrease in soil pH than maize (Figure 2B). It should be noted that the pH of each low-chromium-concentration soil treatment was significantly lower than that of the corresponding high-chromium-concentration soil treatment after adding earthworms (*p* < 0.05) (Figure 2). The addition of earthworms and plants also had a significant effect on the soil organic matter content (Figure 3). In acidic soil, the organic matter content of the earthworm treatment group was significantly lower than that of the group without earthworms (*p* < 0.05) (Figure 3A). In alkaline soil, the organic matter content of the earthworm treatment group was significantly higher than that of the group without earthworms (*p* < 0.05) (Figure 3B).

Furthermore, the removal rate of chromium in the soil was significantly positively correlated with pH and significantly negatively correlated with organic matter content in both the acidic and alkaline soils (Table 3).

Planting ryegrass in low-chromium-concentration soil effectively reduced the concentrations of four different forms of chromium, while planting maize significantly increased the contents of acid-extractable and reducible chromium in the soil (*p* < 0.05) (Figure 4A,B). In high-chromium-concentration soil, growing both ryegrass and maize significantly promoted the content of oxidizable chromium in the soil (*p* < 0.05) (Figure 4C,D). After introducing earthworms, the contents of acid-extractable and reducible chromium in the soil increased significantly, while the contents of oxidizable and residual chromium decreased significantly (*p* < 0.05). The combined ryegrass and earthworm treatment significantly reduced the contents of acid-extractable and reducible chromium in the soil.

Overall, the combination of *E. fetida* and ryegrass had the highest removal rate for chromium in the soil. Earthworms and plants had different significant effects on the pH and organic matter in acidic and alkaline soil. Earthworms effectively enhanced the bioavailability of chromium in the soil, while ryegrass effectively removed active chromium from the soil.

### 3.2. Accumulation of Chromium in Organisms

*Eisenia fetida* exhibited higher chromium accumulation than *P. guillelmi* in both the acidic and alkaline soils (Figure 5). The growth of both of the earthworm species appeared to be inhibited and there was a significant reduction in biomass under high chromium concentrations, but their biomass did not significantly differ at the same chromium concentrations (Appendix A). High chromium concentrations in both soil types also increased the chromium content in both of the earthworm species, with *E. fetida* having higher chromium content than *P. guillelmi* (Appendix A). *Eisenia fetida* showed better adaptability and enrichment ability for chromium than *P. guillelmi*. The chromium accumulation in crops significantly increased after earthworm treatment (*p* < 0.05), except for maize in alkaline soil, which showed varying degrees of decline under high chromium concentrations (Figure 6). The chromium accumulation in ryegrass after *E. fetida* treatment was significantly higher than that of *P. guillelmi* treatment (*p* < 0.05), while there was no significant difference in maize following treatment with either of the earthworm species. Ryegrass had higher chromium accumulation than maize, and high chromium concentrations significantly increased the chromium accumulation in ryegrass (*p* < 0.05). Earthworms can significantly increase the biomass and chromium concentration of plants, but both of the plants had varying degrees of biomass reduction under high-chromium treatments, with ryegrass having significantly higher biomass than maize (Appendix A). The chromium concentration in ryegrass significantly increased under high-soil-chromium concentrations (*p* < 0.05), while the chromium concentration in maize did not increase significantly and even showed a downward trend in varying degrees (Appendix A).

### 3.3. Gut Microbiota of Earthworms

The bacterial diversity at the genus level of both of the earthworm species significantly decreased, especially in acidic soil (Appendix A). Principal component analysis (PCA) showed that the explanatory rates of PC1 and PC2 for the bacterial community composition at the phylum level were 52.67% and 14.95%, respectively, and at the genus level, the values were 42.81% and 14.84%, respectively (Figure 7). The distance between the earthworm control and treatment groups was large, especially between the acidic and alkaline soils. Permutational multivariate ANOVA (PERMANOVA) showed that the earthworm species had a small effect on gut bacterial composition at the phylum level and a large effect at the genus level; soil pH had a significant effect on community composition at the genus level (*p* = 0.031); and chromium concentration had a minor effect on community composition at the genus level (Table 4). The gut bacterial community compositions of the two earthworm species in acidic and alkaline soils were additionally analyzed. Ternary plots at the genus level for both of the earthworm species in acidic and alkaline soils showed significant differences in the relative abundances of gut bacterial communities, with significant increases in the proportions of the top 10 dominant bacterial genera. *Eisenia fetida* and *P. guillelmi* increased their dominant bacterial genera in acidic soil by 26.51% and 23.32%, respectively, and in alkaline soil, they increased their dominant bacterial genera by 20.95% and 11.78%, respectively (Figure 8A–C).

Linear discriminant analysis effect size (LEfSe) analysis screened out 219 microbial genera with significant differences in abundance in alkaline and acidic soils when the linear discriminant analysis (LDA) threshold was 2 (see Appendix A). When the LDA threshold was raised to 4 to more accurately identify genera with significant differences, 11 genera in alkaline soil, including p_*Actinobacteriota* and c_*Actinobacteria*, were significantly enriched (Figure 8D).

Detrended correspondence analysis (DCA) showed a unimodal nonlinear relationship between enzyme activity and bacterial community abundance. Canonical correlation analysis (CCA) showed that MDA, SOD, and GSH-Px were negatively correlated with the gut bacterial community composition of earthworms to varying degrees, while GST and CAT were positively correlated, and ROS was not significantly correlated (Figure 9A). The correlation heatmap showed that MDA was significantly negatively correlated with 14 bacterial genera, while SOD was significantly negatively correlated with 11 bacterial genera and significantly positively correlated with *Ralstonia*. GST and GSH-Px were significantly positively correlated with *Ralstonia*, unclassified_f__*Microbacteriaceae*, and *Verminephrobacter*, and were significantly negatively correlated with *Citrobacter* and unclassified_o_*Enterobacterales*. CAT was significantly positively correlated with *Candidatus_Berkiella* and was significantly negatively correlated with unclassified_f__*Microbacteriaceae* (Figure 9B).

## 4. Discussion

### 4.1. Effects of Earthworms and Plants on Chromium Removal

The removal rate of chromium from soil is an important indicator of the effectiveness of environmental remediation. In the present study, it was found that the application of earthworms and plants together significantly improved the removal rate of chromium from soil in most treatments, with *E. fetida* and ryegrass showing the best ability for the synergistic removal of chromium. However, in alkaline soil, the synergistic removal of chromium by earthworms and maize was not significantly improved. This may be due to the narrow range of maize adaptation to soil pH and also because alkaline soil contains excessive alkaline ions such as sodium and calcium, which can have negative effects on maize nutrient absorption and growth [40].

The speciation of heavy metals in soil changes under different pH conditions, affecting the phytoremediation of heavy-metal-polluted soil [41]. Earthworms have a bidirectional effect on soil pH, with acidic and alkaline soils tending to become neutral after ingestion [42,43,44]. Previous studies have shown a negative correlation between soil pH and heavy metal bioavailability [45,46]. Interestingly, increasing the pH in acidic soil in the present study improved the chromium removal efficiency, indicating that the effect of earthworms on heavy metal bioavailability was not solely due to changes in soil pH caused by earthworms and that the original soil acidity and alkalinity also played a significant role. Organic matter is important for maintaining soil health and fertility and can form complexes with heavy metals, thereby reducing their biological availability [47,48,49]. The introduction of earthworms led to significant differences in organic matter content changes in acidic and alkaline soils. Increasing the soil organic matter content can reduce the plant uptake of heavy metals, affecting the heavy metal removal efficiency. Increasing the organic matter content in alkaline soil in the present study significantly increased the chromium removal efficiency, similar to the effect of soil pH, highlighting the complex relationship between earthworms, soil acidity and alkalinity, and heavy metal bioavailability.

Soil heavy metals can be divided into four forms: acid-extractable, reducible, oxidizable, and residual forms. Among them, the acid-extractable and reducible forms have higher bioavailability and can be absorbed and utilized by plant roots, while the oxidizable form has relatively lower bioavailability, and the residual form exists in the solid part of the soil as a potential nonflowing soil component [50]. In the present study, the addition of earthworms to the soil significantly increased the contents of acid-extractable and reducible chromium, while significantly reducing the contents of oxidizable and residual chromium compared to the treatment with plants alone. This result indicates that earthworms can effectively increase the bioavailability of chromium in soil and activate chromium in soil, which is consistent with the research results of Sizmur et al. [51,52].

Overall, the results indicate that via the synergistic effect of earthworms and suitable plants, the removal rate of chromium in soil can be increased, and the occurrence form of heavy metals in soil can be changed, increasing their bioavailability. In addition, the effect of earthworms on soil pH is bidirectional and can promote the decomposition and accumulation of organic matter, thereby maintaining the health of the soil ecosystem.

### 4.2. Scavenging of Chromium by Earthworms and Plants

The enrichment of chromium by earthworms and plants reflects their feasibility and effectiveness as a potential means of bioremediation for heavy metal pollution. In the present study, *E. fetida* showed significantly higher biomass, chromium concentration, and accumulation than *P. guillelmi*. This indicates that *E. fetida* has stronger adaptability and accumulation ability in response to chromium-polluted soil. In addition, at high chromium concentrations, the biomass of both of the earthworm species decreased significantly, consistent with the findings of Fernando et al. [53], who found that tissue changes occurred in earthworms exposed to soil with Cr concentrations ranging from 0.24 to 893 mg/kg, including cell fusion, reduced epidermal thickness, and epidermal shedding, even at the lowest concentration. Gupta et al. also found that chromium-contaminated soil had a significant impact on earthworm reproduction [54].

The efficiency of the phytoremediation of heavy metal pollution depends on plant biomass and the concentration of heavy metals in plants [55]. In this experiment, the co-removal of chromium by earthworms and plants significantly increased the biomass of ryegrass and maize and the chromium concentration in plant tissues, promoting chromium accumulation. Similarly, Liu et al. found that earthworm activity significantly increased the contents of available nitrogen and phosphorus in sludge applied as fertilizer, thereby increasing the biomass of cabbage [56]. In addition, other studies have shown that earthworm activity can increase the abundance of soil microbial communities, thereby promoting the growth of tomato seedlings [57]. These results suggest that the presence of earthworms can improve plant growth and nutritional status, while earthworm activity can also promote microbial growth, thereby accelerating the ability of plants to remediate heavy metals from contaminated soils. Moreover, studies have shown that earthworm secretions and castings can change the form of heavy metals in soil and increase their bioavailability, thereby increasing the absorption of heavy metals by plants. For example, Zhang et al. found that adding earthworm secretions increased the concentration and accumulation of Cd in plants [58]. The addition of earthworm castings increased the content of heavy metals in clover, while the contents of Fe, Cu, and Cr in the soil extracted using Diethylenetriaminepentaacetic acid (DTPA) decreased [59].

In summary, treatment with high chromium concentrations resulted in some inhibition of earthworm and plant growth. The introduction of earthworms significantly increased the efficiency of the phytoremediation of heavy metals.

### 4.3. Responses of Earthworm Gut Microbial Communities

The gut microbiota of earthworms play a crucial role in regulating the plant remediation of contaminated soil via microbial metabolism and adsorption effects [60]. Therefore, the analysis of earthworm gut microbiota is of great significance for the remediation of heavy-metal-polluted soil. However, the diversity and abundance of earthworm gut microbiota are affected by many environmental factors. Boughattas et al. found that in the background of heavy metal pollution, the influence on the bacterial community structure changed and the health of earthworms was significantly affected [61]. Other studies have shown that cadmium exposure disrupts the balance of earthworm gut bacterial communities and increases the abundance of heavy-metal-resistant bacteria [62]. These suggest that the earthworm gut microbiota respond to metal pollution in the surrounding soil.

Comparing the microbial community composition under different treatments revealed significant differences in the microbial community between earthworms fed chromium-contaminated soil samples and control samples, indicating an impact of earthworm feeding on the gut microbial community. This is similar to the findings of Li et al. [63]. The dominant bacteria at the phylum level, including *Proteobacteria*, *Actinobacteria*, and *Firmicutes*, showed significant changes in abundance and composition, while the relative abundance of the bacterial community at the genus level varied greatly. PERMANOVA showed that although chromium pollution had an impact on the gut microbiota, there was no significant difference between high and low concentrations of chromium. The gut microbiota of earthworms showed significant differences between acidic and alkaline soils, and the species diversity in the gut microbiota with alkaline soil was richer. The reason for this may be that the pH of the earthworm gut environment is usually in the neutral to weakly alkaline range. The earthworms had higher chromium accumulation in acidic soil, which may have caused greater stress to the earthworms [64]. Hait and Tare’s research also showed that environmental factors, including pH, temperature, humidity, oxygen content, light, and organic matter in heavy-metal-contaminated soil, had an impact on the growth and survival of earthworms [65]. Therefore, the decrease in microbial diversity may be the result of the comprehensive effects of environmental conditions and microbial metabolic activities. Via LEfSe differential analysis, 11 different bacterial genera in alkaline soil were screened out, belonging to *Bacillales*, *Alphaproteobacteria*, and *Actinobacteriota*. These genera may have special metabolic pathways and enzyme systems required to adapt to alkaline conditions and can survive in a high-pH environment. In addition, studies have shown that adding *Bacillales* strains to cadmium-contaminated soil can increase the mobility and bioavailability of cadmium, resulting in a two-fold increase in the plant uptake of cadmium [66]. In alkaline soil containing chromium and other heavy metals, these microorganisms may also play a role in the biological remediation of chromium-contaminated soil, reducing the chromium content in soil via metabolic or adsorption effects.

ROS are highly reactive oxidants produced within cells, and excessive ROS can cause damage to organisms and induce oxidative stress responses [67]. Antioxidant enzymes, including SOD, CAT, and GST, play critical roles in eliminating excess ROS within organisms [68,69]. MDA is an important product of lipid peroxidation and is considered to be a reliable biomarker for evaluating the level of lipid peroxidation [70]. Studies have shown that there is a significant correlation between enzyme activity in the body of earthworms and their intestinal microbiota [71,72]. In the present experiment, MDA, SOD enzyme, and bacteria such as *Pseudomonas* and *Bacillus* were significantly negatively correlated. *Bacillus* is resistant to heavy metals and has the ability to remediate heavy metal pollution [73]. It has also been found that groups of bacteria such as *Pseudomonas*, *Enterobacter*, and *Chromobacterium* act as metal carriers in soil and can interact with heavy metals via specific chemical reactions to form stable metal compounds, and then remove them from the soil [74,75]. GST and GSH-Px enzymes were significantly positively correlated with the *Verminephrobacter* community in the present study. *Verminephrobacter* is a type of bacterium that can symbiotically live in the gut of earthworms. When earthworms are exposed to environmental stress (such as high temperature and drought), the quantity and population structure of *Verminephrobacter* within their bodies change, thereby affecting the earthworm’s tolerance [76]. This study found a significant correlation between changes in enzyme activity and the intestinal bacterial community of earthworms following ingesting chromium-containing soil, which was similar to previous research results [77]. In addition, the high abundances of *Chryseobacterium*, *Citrobacter*, and unclassified *Enterobacterales* in acidic chromium-containing soil may have been due to their unique physiological characteristics. For example, Li et al. isolated and identified chromium-resistant bacteria from tannery wastewater and found that *Chryseobacterium* had strong acid and chromium resistance and enhanced the biological activity of chromium [78].

## 5. Conclusions

This study found that *E. fetida* had a higher removal rate of Cr from soil than *P. guillelmi*, and ryegrass was more effective in removing Cr than maize. The most efficient removal of Cr from low-concentration acidic soil was achieved via the synergistic effect of *E. fetida* and ryegrass, with a removal rate of 62.64%. Earthworms increased the bioavailability of heavy metals in soil, but their efficacy varied under different pH conditions. Metal accumulation analysis showed that the concentration of Cr in earthworms was higher in acidic soil than in alkaline soil. After gut passage, earthworms significantly increased the bioavailability of Cr in soil, promoting its accumulation in plants. The characteristics of earthworm gut microbial communities were significantly correlated with soil pH and were affected by Cr-contaminated soil. Certain bacterial strains such as *Bacillales* in alkaline soil showed resistance to Cr. The bacterial groups *Pseudomonas* and *Verminephrobacter* were found to be closely related to the bioavailability of Cr in soil and the stress imposed on earthworms due to Cr exposure. *Chryseobacterium* and *Citrobacter* may have strong abilities to tolerate acidic soil and Cr and enhance the bioavailability of Cr. The bioavailability of heavy metals in plants is closely related to bacterial activity, and earthworms may act as a bridge between heavy metals and plants. As a part of the heavy metal amendment of soil, earthworms and plant residues that have been enriched for heavy metals must be removed from soil for the purpose of soil remediation. Future research will focus on screening more hyperaccumulator plants and animals, which can be combined with modern molecular biology techniques and transgenic technologies to produce more eco-friendly varieties with enhanced stress tolerance, to evaluate their synergistic efficiency for the removal of heavy metals from soil and to promote the health of soil ecosystems.

## Figures and Tables

**Figure 1 biology-12-00831-f001:**
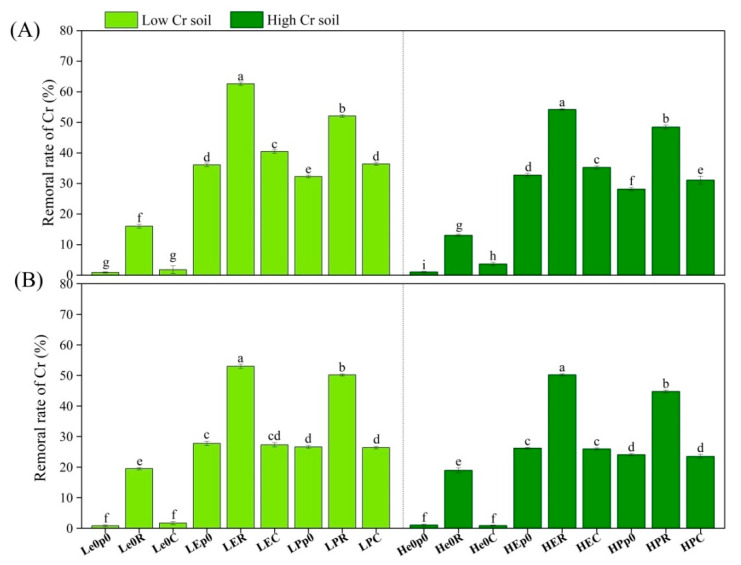
Removal rate of Cr in soil. Removal rate of Cr at 15 mg/kg and 50 mg/kg in acidic soil (**A**) and alkaline soil (**B**) after synergistic removal by earthworms and plants. Values are described as mean ± SD (*n* = 3). Different lowercase letters above columns indicate significant differences at *p* < 0.05 level among treatments. Note: L, low Cr content; H, high Cr content; e0, no earthworm inoculation; E, inoculation of *E. fetida*; P, inoculation of *P. guillelmi*; p0: no plants; R: planting ryegrass; C: planting maize; the same is for below.

**Figure 2 biology-12-00831-f002:**
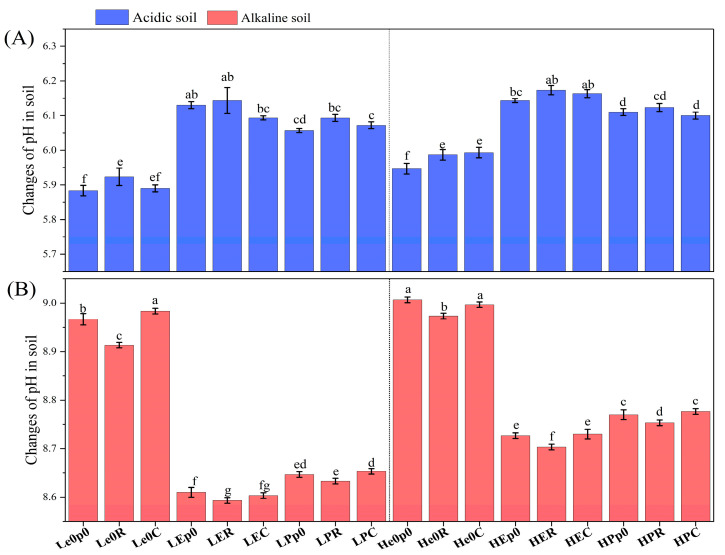
Changes in pH in soil. Changes in pH at 15 mg/kg and 50 mg/kg in acidic soil (**A**) and alkaline soil (**B**) after synergistic removal by earthworms and plants. Values are described as mean ± SD (*n* = 3). Different lowercase letters above columns indicate significant differences at *p* < 0.05 level among treatments.

**Figure 3 biology-12-00831-f003:**
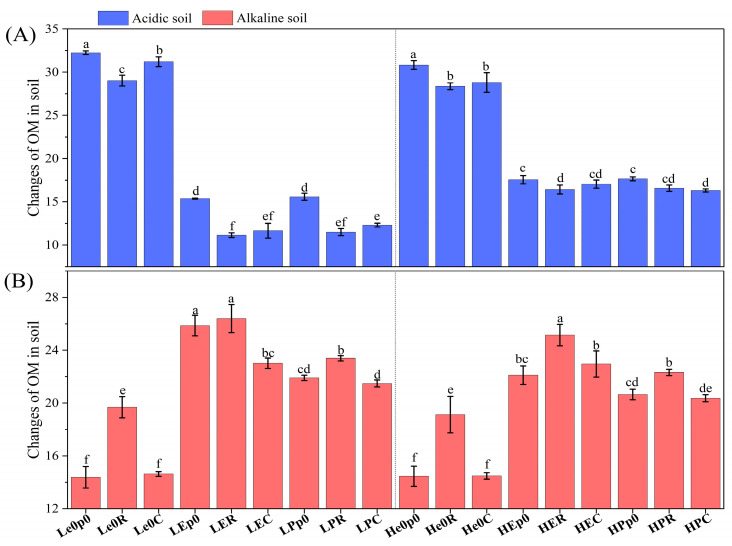
Changes in OM in soil. Changes in OM at 15 mg/kg and 50 mg/kg in acidic soil (**A**) and alkaline soil (**B**) after synergistic removal by earthworms and plants. Values are described as mean ± SD (*n* = 3). Different lowercase letters above columns indicate significant differences at *p* < 0.05 level among treatments.

**Figure 4 biology-12-00831-f004:**
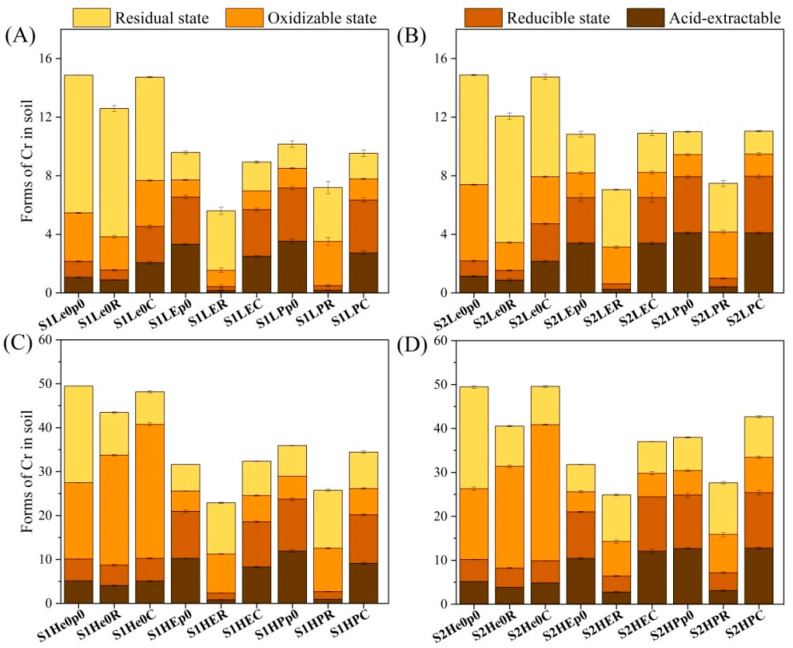
Changes in various forms of Cr in the soil. Forms of Cr in 15 mg/kg and 50 mg/kg acidic soil (**A**,**C**) and alkaline soil (**B**,**D**) after synergistic removal by earthworms and plants. Values are described as mean ± SD (*n* = 3). Note: S1, acidic soil; S2, alkaline soil.

**Figure 5 biology-12-00831-f005:**
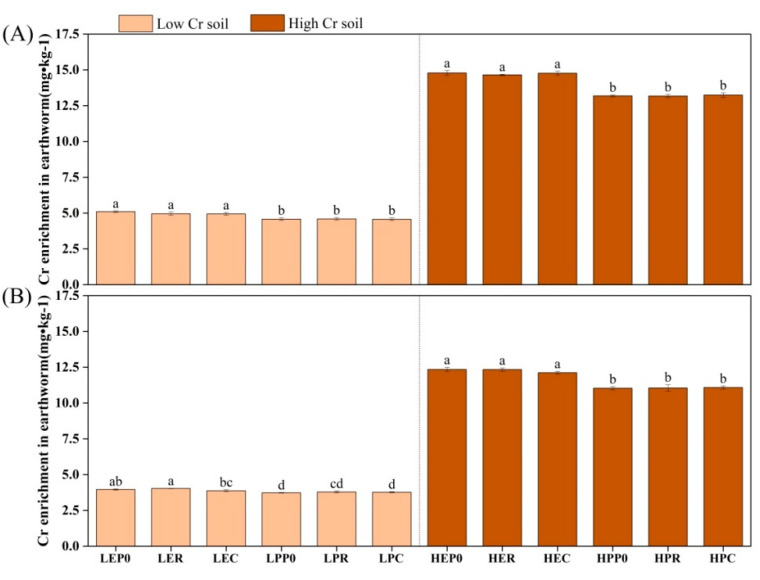
The amount of Cr enrichment in earthworms. The amount of Cr enrichment in earthworms at 15 mg/kg and 50 mg/kg in acidic soil (**A**) and alkaline soil (**B**). Values are described as mean ± SD (*n* = 3). Different lowercase letters above columns indicate significant differences at *p* < 0.05 level among treatments.

**Figure 6 biology-12-00831-f006:**
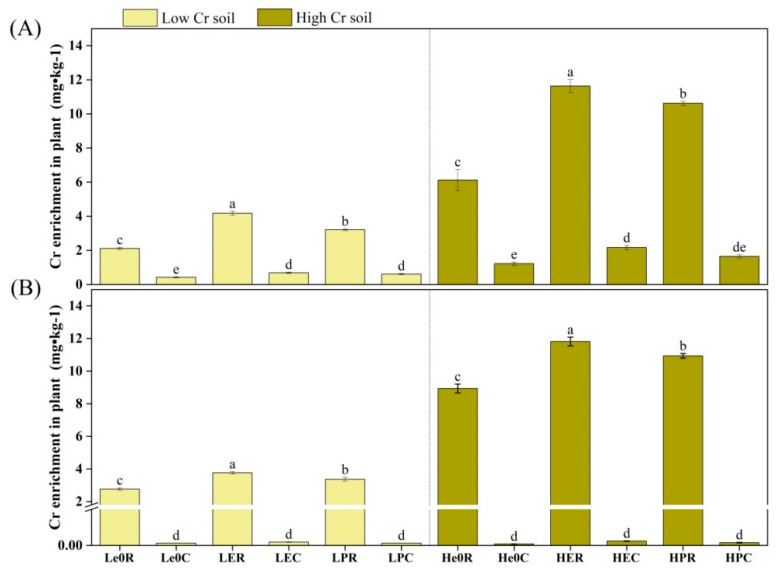
The amount of Cr enrichment in plants. The amount of Cr enrichment in plants at 15 mg/kg and 50 mg/kg in acidic soil (**A**) and alkaline soil (**B**) after removal by earthworms. Values are described as mean ± SD (*n* = 3). Different lowercase letters above columns indicate significant differences at *p* < 0.05 level among treatments.

**Figure 7 biology-12-00831-f007:**
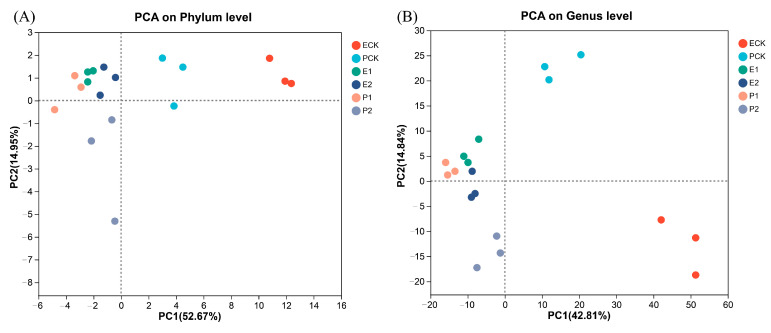
Principal component analysis of species composition among intestinal bacterial communities at the phylum level (**A**) and genus level (**B**) of earthworms. Abbreviations: ECK, PCK (primitive *E. fetida* and *P. guillelmi*); E1, E2 (*E. fetida* in acidic and alkaline soils); P1, P2 (*P. guillelmi* in acidic and alkaline soils).

**Figure 8 biology-12-00831-f008:**
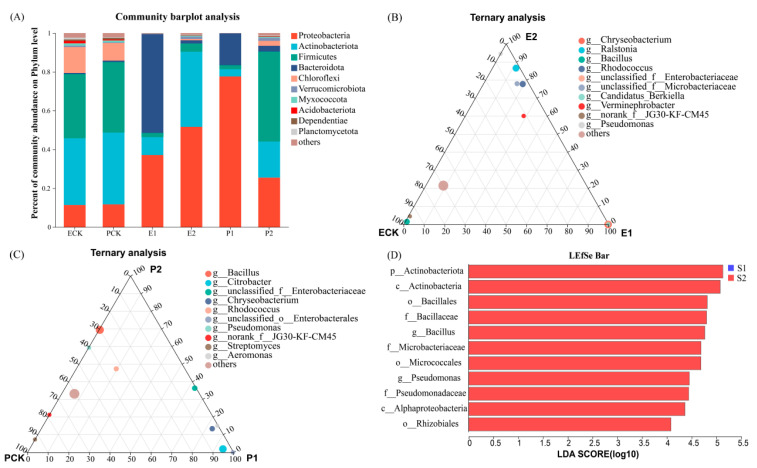
Composition and abundance characteristics of bacterial communities at the phylum level of two earthworms in acidic and alkaline soils (**A**), and composition and abundance characteristics of bacterial communities at the genus level of *E. fetida* (**B**) and *P. guillelmi* (**C**). Differentiated taxa between groups based on analysis results of LEfSe (with LDA score set at 4) (**D**).

**Figure 9 biology-12-00831-f009:**
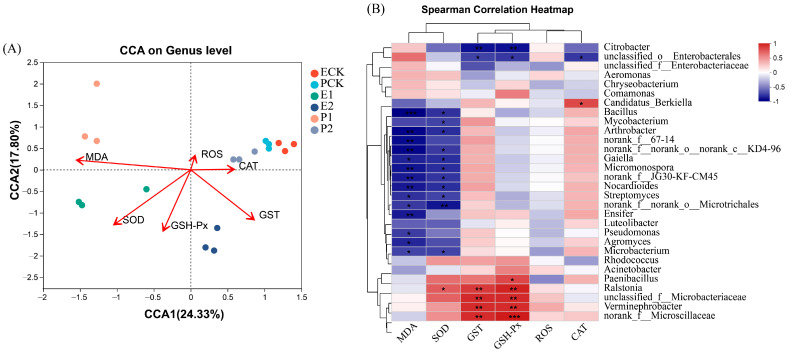
CCA analysis of bacterial community at the genus level and enzyme activity in the intestinal tract of earthworms (**A**), and correlation heat map (**B**). *, significant correlation, **, highly significant correlation, ***, extremely high correlation.

**Table 1 biology-12-00831-t001:** Properties of the soil used for the experiment.

Soil Properties	Acidic Soil	Alkaline Soil
pH value	5.88 ± 0.06	8.97 ± 0.12
Organic matter Cr (mg kg^−1^)	32.25 ± 0.36	14.38 ± 0.14
Acid-extractable Cr (mg kg^−1^)	1.13 ± 0.08	1.16 ± 0.10
Reducible Cr (mg kg^−1^)	1.12 ± 0.03	1.07 ± 0.04
Oxidizable Cr (mg kg^−1^)	3.37 ± 0.10	5.24 ± 0.13
Residual Cr (mg kg^−1^)	9.55 ± 0.08	7.49 ± 0.09
Total Cr content (mg kg^−1^)	15.17 ± 0.33	14.96 ± 0.14

**Table 2 biology-12-00831-t002:** Experimental design for ecotoxicity test: (a) presentation of different factors and treatments, (b) description of treatments with different combinations of factors. The experimental design involved four factors: (i) two acidic and alkaline soils (S1, S2), (ii) two chromium concentrations (L, H), (iii) earthworms (E, P), and (iv) plants (R, C) to create a set of different combinations of the earthworms and plants present in acidic or alkaline soils contaminated with different concentrations of chromium, with each treatment repeated three times.

(a) Factors		Treatments	
Soils		Acidic soil (S1)	
		Alkaline soil (S2)	
Concentration		15 mg/kg (L), 50 mg/kg (H)	
Earthworms		No earthworms (e0)	
		*E. fetida* (E)	
		*P. guillelmi* (P)	
Plants		No plants (p0)	
		Ryegrass (R)	
		Maize (C)	
**(b) Treatment**	**Description**	**Treatment**	**Description**
1	S1Le0p0	19	S2Le0p0
2	S1Le0R	20	S2Le0R
3	S1Le0C	21	S2Le0C
4	S1LEp0	22	S2LEp0
5	S1LER	23	S2LER
6	S1LEC	24	S2LEC
7	S1LPp0	25	S2LPp0
8	S1LPR	26	S2LPR
9	S1LPC	27	S2LPC
10	S1He0p0	28	S2He0p0
11	S1He0R	29	S2He0R
12	S1He0C	30	S2He0C
13	S1HEp0	31	S2HEp0
14	S1HER	32	S2HER
15	S1HEC	33	S2HEC
16	S1HPp0	34	S2HPp0
17	S1HPR	35	S2HPR
18	S1HPC	36	S2HPC

**Table 3 biology-12-00831-t003:** Correlation between chromium removal rate and soil physical and chemical properties.

Soil Index	S1-L	S1-H	S2-L	S2-H
pH	0.856 **	0.868 **	−0.892 **	−0.720 **
OM	−0.949 **	−0.869 **	0.929 **	0.763 **

Note: **, highly significant correlation.

**Table 4 biology-12-00831-t004:** PERMANOVA on the effects of three different experimental factors on bacterial community composition.

Taxonomy	Characteristics	Sums of Sqs	Mean Sqs	F_Model	R^2^	P_Adjust
Phylum	E-P	0.100	0.100	0.542	0.083	0.740
Genus	E-P	0.589	0.589	1.992	0.249	0.060
Phylum	S1-S2	0.353	0.354	2.472	0.293	0.070
Genus	S1-S2	0.681	0.681	2.433	0.289	0.031
Phylum	H-L	0.228	0.228	1.396	0.189	0.261
Genus	H-L	0.253	0.253	0.721	0.107	0.730

(E: *E. fetida*; P: *P. guillelmi*; S1: acidic soil; S2: alkaline soil; H: high chromium; L: low chromium).

## Data Availability

The data used to support the findings of this study are available from the corresponding author upon request.

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
