# Peer review of "Synergistic Effects of Earthworms and Plants on Chromium Removal from Acidic and Alkaline Soils: Biological Responses and Implications"

_biology, 2023, doi:10.3390/biology12060831_

Round 1
Reviewer 1 Report
I must express that I find the manuscript to be of great scientific value, and I have no specific feedback to provide at this time.
Author Response
Thanks for reviewer 1's kindly review, and there is no comments and responses.
Reviewer 2 Report
This draft describes a complex factorial experiment involving two soil types, three earthwork treatments, three plant treatments imposed on two levels of chromium additions in three replications of a pot experiment to evaluate the impact of the 36 treatments on chromium removal from soil. The paper also examines the impact of the treatments on the gut microflora of the two earthworm species evaluated. The study appears to have been meticulously carried out.
There are some limitations in the project as described, some attributable to the relatively recent interest in potential heavy metal remediation of soils using, among other attributes, earthworms. The paper is expressed in excellent English, and acknowledges linguistic assistance having been provided in preparing the paper. However, there is some evidence that in ensuring polished English expression, some of the original text may have become obscured and the linguistic adviser has not understood the science..
The following comments are made
Line 65-67 – This is a crucial statement, and the authors need to identify in their paper when they are dealing with contaminated soil or polluted soil, in the latter case the growth of organisms will be slowed/inhibited.
Line 106 – The authors should identify how their chosen Chromium concentrations are regarded in comparison with other authors’ definitions of contaminated or polluted levels of Chromium.
Lines 115-117 do not describe correctly the implementation of the intended treatments. The text reads “In addition, 40 E. fetida and 20 P. guillelmi were added to every pot”, but presumably E. fetida were added to only those pots containing that treatment, not to every pot. Likewise for P. guillelmi.
Table 2 is unsatisfactory. The first two columns list the variables line by line, except in the case of Chromium concentration, where two variables are shown on one line. The third and fourth columns should be restructured into a new table. The 9 listed treatments are not congruent with the 36 actual treatments which the experiment contains. The descriptions provided are also confusing as they contain several alternative treatments in beach line.
Line 143 – The method of collection of earthworm samples from the soil should be defined.
Line 168 – The authors should identify why only the plant tops were harvested, noting whether Chromium accumulation may have also been anticipated in the roots. (There is discussion later at line 357 about root accumulation.)
Line 172 refers to Chromium in the “plant tissue” but presumably the “harvested tissue” is what is meant.
Figure 1 (and subsequent figures) – The column abbreviations on the “x” axis are not identified – it is left for the reader to work them out. Perhaps they could be incorporated into a revised version of the two right-hand columns of the current table 2.
Figures 2 and 3 – These are shown as graphs linking the treatment points. This form of expression is not suitable as the increments on the “x” axis are not continuously variable parameter measures. Histograms should be used as has been done in the other figures.
Figures 4, 5 and 6 – Again, the nomenclature on the “x” axes is undefined.
Line 330-331 states “it was found that the application of earthworms and plants together significantly improved the removal rate of chromium from soil,” yet subsequently lines 333-334 partly contradict the statement, observing “:the synergistic removal of chromium by earthworms and maize was not significantly improved.”
Line 373 refers to “The enrichment of chromium in soil by earthworms and plants” But the paper is about enrichment of chromium in earthworms and plants, and its reduction in soils. Confusing?
Line 402 – Again, the paper states “In summary, earthworms not only significantly enrich chromium in soil…” which is presumably not what the authors are concluding.
Line 456 states “Verminephrobacter is a type of bacteria that can symbiotically live in the kidney cells of earthworms.” Earthworms do not have kidneys. Earthworms have nephridia to filter out the dead cells and other wastes that are sloughed into the blood. Wastes from the nephridia are eliminated through the same opening as the digestive wastes.
The Conclusion suggests that earthworms increase bioavailability of chromium so it can be more readily be taken up by plants, and is hence removed from the soil. However, the authors do not discuss whether plant residues and earthworm residues have to be removed from the soil to ensure the chromium soil remediation. Some authors specify that worms and their residues should be removed from the soil as part of the amelioration process. The authors should comment on this if they can.
The English is well expressed, but there may be places where what is expressed is not what the authors may have intended, notably with regard to where chromium is concentrated by the treatments.
Reviewer 3 Report
I read the manuscript Synergistic Effects of Earthworms and Plants on Chromium Removal from Acidic and Alkaline Soil: Biological Responses and Implications, by Ping Liu , Yan Song , Jie Wei , Wei Mao , Jing Ju , Shengyang Zheng , Haitao Zhao, with a great interest.
The subject of the research has a great practical impact. The authors made huge job, cunducting all these experiments and preparing results.
I definitely recommend this manuscript for publication after major revision.
However, I have several comments:
- Abstract section should have been rewritten. Please, add several introductory sentences. Why did you do all these experiments. Your Introduction section is well prepared, so you can easily take some points from this section and include in Abstract.
-In materials and methods section please write how did you study gut microbiota.
- Results section should have been rewritten:
- present means and sd in text,
-define X-axis on graphs,
-please specify and define all abbreviations on X-axis (LeOR, LeOC, LPC....). I'm not sure that most readers will look at additional files.
-maybe it's better to reorganise your data on Fig 1-3, 5-6 to make it easier to compare.
- I do not understand what do all lowercase letters mean in Fig 1-3, 5-6. What the difference between a and b or a and d? Please make it more reader-friendly.
-please cite not Figure X, but Figure XA or B, specify each graph on figure.
That's all my comments.
Overall, I think authors made a great job, however, they need improve the manuscript presentation according to comments.
Reviewer 4 Report
Please find comments from the attached file

The quality of English is average, and can be improved by a good/professional English editor
Round 2
Reviewer 2 Report
The authors have addressed the various comments made on the original version. My only remaining concern is t the sentence "A 30 g/L solution of potassium dichromate was added to both soil types and stirred evenly to prepare the contaminated soil" (line 119) , but I assume it was only added to the two soil types covering the High 50 mg/kg treatment as the Low treatment soils were already at 15.17 and 14.96 mg/kg (Table 1) respectively. If so, the text could be slightly further clarified. I also felt the sentence (line 185) reading "After attaining a desired growth, the above ground parts of the plants were harvested..." is rather vague, whatever "desired growth" means. Otherwise , the draft is looking good.
Author Response
Response to Reviewer 2Comments
Point1: The authors have addressed the various comments made on the original version. My only remaining concern is t the sentence "A 30 g/L solution of potassium dichromate was added to both soil types and stirred evenly to prepare the contaminated soil" (line 119), but I assume it was only added to the two soil types covering the High 50 mg/kg treatment as the Low treatment soils were already at 15.17 and 14.96 mg/kg (Table 1) respectively. If so, the text could be slightly further clarified.
Response 1: Sure, this sentence is kind of confusing, and have been written as “A 30 g/L solution of potassium dichromate was added to both soil types and stirred evenly to prepare the contaminated soil with high concentrations of chromium (50 mg/kg) while the original chromium concentration in the soils was considered as the low chromium concentration (approximately 15 mg/kg).” See line 131-134.
Point2: I also felt the sentence (line 185) reading "After attaining a desired growth, the above ground parts of the plants were harvested..." is rather vague, whatever "desired growth" means. Otherwise, the draft is looking good.
Response 2: The sentence has been replaced with “When the plants were mature,” See line 199.
Reviewer 3 Report
Dear authors, you've made a great job and meet all the comments, thnk you for that. Now, I hust have one comment:
lines 479-480: It has also been found that groups of bacteria such as Pseudomonas, Enterobacter and Chromobacterium act as metal carriers in the soil and can interact with ...
use Italics
Now I recommend accept manuscript in present form and best wishes to authors!

Author Response
Response to Reviewer 3Comments
Dear authors, you've made a great job and meet all the comments, thank you for that. Now, I must have one comment:
Point1:lines 479-480: It has also been found that groups of bacteria such as Pseudomonas, Enterobacter and Chromobacterium act as metal carriers in the soil and can interact with ...
use Italics
Response 1: Pseudomonas, Enterobacter and Chromobacterium are Italics now in line 493-494.
Reviewer 4 Report
Authors improved their manuscript
Author Response
Thanks for your review. There are on comments and responses